# A Weighted-Time-Lag Method to Detect Lag Vegetation Response to Climate Variation: A Case Study in Loess Plateau, China, 1982–2013

Qianqian Sun [1], Chao Liu [1,2,*] , Tianyang Chen [3] and Anbing Zhang [2]

1   School of Spatial Informatics and Geomatics Engineering, Anhui University of Science and Technology, Huainan 232001, China; 2018200913@aust.edu.cn
2   School of Mining and Geomatics, Hebei University of Engineering, Handan 056038, China; zhanganbing@hebeu.edu.cn
3   Department of Geography and Earth Science, The University of North Carolina at Charlotte, Charlotte, NC 28223, USA; tchen19@uncc.edu
*   Correspondence: liuchao0202@hebeu.edu.cn; Tel.: +86-182-5542-0251

**Abstract:** Vegetation fluctuation is sensitive to climate change, and this response exhibits a time lag. Traditionally, scholars estimated this lag effect by considering the immediate prior lag (e.g., where vegetation in the current month is impacted by the climate in a certain prior month) or the lag accumulation (e.g., where vegetation in the current month is impacted by the last several months). The essence of these two methods is that vegetation growth is impacted by climate conditions in the prior period or several consecutive previous periods, which fails to consider the different impacts coming from each of those prior periods. Therefore, this study proposed a new approach, the weighted time-lag method, in detecting the lag effect of climate conditions coming from different prior periods. Essentially, the new method is a generalized extension of the lag-accumulation method. However, the new method detects how many prior periods need to be considered and, most importantly, the differentiated climate impact on vegetation growth in each of the determined prior periods. We tested the performance of the new method in the Loess Plateau by comparing various lag detection methods by using the linear model between the climate factors and the normalized difference vegetation index (NDVI). The case study confirmed four main findings: (1) the response of vegetation growth exhibits time lag to both precipitation and temperature; (2) there are apparent differences in the time lag effect detected by various methods, but the weighted time-lag method produced the highest determination coefficient ($R^2$) in the linear model and provided the most specific lag pattern over the determined prior periods; (3) the vegetation growth is most sensitive to climate factors in the current month and the last month in the Loess Plateau but reflects a varied of responses to other prior months; and (4) the impact of temperature on vegetation growth is higher than that of precipitation. The new method provides a much more precise detection of the lag effect of climate change on vegetation growth and makes a smart decision about soil conservation and ecological restoration after severe climate events, such as long-lasting drought or flooding.

**Keywords:** climate change; precipitation; temperature; NDVI; time-lag; Loess Plateau

## 1. Introduction

In recent decades, global climate change has attracted much attention since it has been assumed to significantly impact territory ecosystems [1–5]. Ecosystem services produced by such ecosystems are essential to social activities on the earth's surface. Therefore, the effect of the climate on ecosystem services has become a boundary object that many researchers from different study areas have collaboratively worked on, including but not limited to geography, ecology, climatology, and biology. Much research has contributed to revealing the impact of different climate factors on vegetation ecosystems in various spatial and temporal scales [6–11].

The lag response of vegetation to the climate in time series has been detected in many studies [12–15]. Such a lag exhibits differences across study regions and vegetation communities [16]. The differences were detected in the impact of precipitation on grassland in the Central Great Plains, USA, and Central Asia [17,18]. Braswell et al. [19] found that different vegetation shows different lag response times to global temperature. The diversity has also been detected in the precipitation and vegetation in Patagonian Grassland by Jobbágy and Sala [20]. This impact also exhibits temporal scales dependency, where it is reported not only on the inter-annual scale but also on seasonal and monthly scales [21,22]. Sala et al. [23] pointed out that the aboveground net primary productivity (ANPP) shows a lag response to the precipitation in the inter-annual scale. Bunting et al. [24] pointed out that shrub-land and woodland have a long response in 6–12 months, and grassland has a shorter response in 3–6 months. Anderson et al. [25] found that the lag responses of the forest to precipitation and sunlight duration were two months and one month, respectively, in the Amazon basin. Kong et al. [26] found that the lag time of normalized difference vegetation index (NDVI) to temperature is not obvious, while the NDVI response lags behind cumulative precipitation by zero to one month, relative humidity by two months, and sunshine duration by three months.

We broadly review the method used to represent the time lag [17,27–29] and proposed typology of paradigms as their essential assumptions, lag, and lag-accumulation to summarize the methods used in the past. To simplify and represent such a process by which vegetation responds to the climate, these methods generate different assumptions. The lag method assumes that the climate factors impact the vegetation in the time (month, year, season, or other scales) of interest in a particular previous month (year, season, or other scales). With such a method, Wu et al. [30] delivered the spatial pattern of the lag response of vegetation to global precipitation, temperature, and sunlight duration. Chuai et al. [29] used this method to detect the lag responses of vegetation communities to precipitation and temperature on a seasonal scale in the Inner Mongolia Plateau. Different from the lag method, lag accumulation assumes that the lags of climate impact are from the previous one or more time periods. With such an assumption, Huang et al. [31] determined the spatiotemporal patterns of climatic effects on global vegetation growth considering various scenarios of time-lag and/or accumulation effects; Richard and Poccard [27] analyzed the spatial patterns of time lag in South Africa and found that the response is not only a lag to a particular month but also responds to a consecutive period. The lag-accumulation method delivers more detailed lag information for the response of the ecosystem to climate factors.

Compared with lag, lag accumulation can better represent a continuous effect process from climate to vegetation, rather than one certain period, where the assumption is that climate factors share different impacts in different months. However, the climate factors in lag months may contribute various weights to vegetation. What is the relative contribution of previous months' climate to explaining vegetation in the current month?

Considering that the impact of climate on vegetation exhibits, consecutively, diversity and accumulation [28,32–35], we propose a weighted time-lag method to address the question. In this method, we consider the ecosystem to have a lag response to climate factors in previous periods, and precipitation and temperature in the previous months affect the vegetation growth in the current month at different weights. The time lag responses of vegetation to climate are detected in order for decision-makers to take action to protect the ecosystem from incoming changes or disturbance. The new method reveals more detailed knowledge about the lag response of the ecosystem to climate.

In this study, we use Loess Plateau, China, as a case study to compare the different results drawn from different lag methods. We have strong background knowledge to verify the method is running as it is supposed to be, and we can validate the results by many case studies addressing such a vulnerable ecosystem in the semi-arid area.

## 2. Study Area and Datasets

### 2.1. Study Area

Loess Plateau (Figure 1), located at the midstream of the Yellow River in China, with an area of 620,000 km² [15], is threatened by drought and desertification due to the uneven and low precipitation and intense evapotranspiration. The precipitation is unevenly distributed in time and space. The annual precipitation from the southeast (800 mm/year) to the northwest (150 mm/year) gradually decreases, while the precipitation in high-flow years is approximately triple or quadruple that in low-flow years. Moreover, the annual average temperature also appears a gradient from the southeast (14.3 °C) to the northwest (4.3 °C) [36].

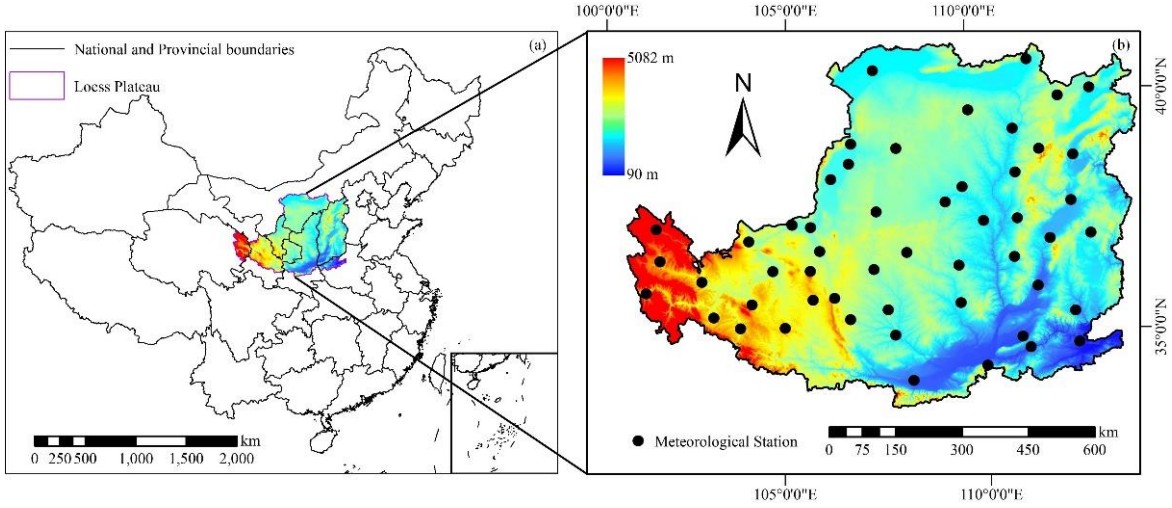

**Figure 1.** The location of the Loess Plateau including elevation and the distribution of meteorological stations.

### 2.2. Datasets

We used NDVI to represent the greenness of vegetation, indicating vegetation growth. NDVI time series (1982–2013), retrieved from the NDVI3g dataset of GIMMS, were collected by NOAA/AVHRR with a temporal resolution of 15 days and a spatial resolution in 1/12 degree (8 km). Furthermore, we conducted pre-processing such as geometry correction, radiance correction, and atmosphere correction [30,37]. Finally, we used maximum value composite to get the monthly NDVI data, which can minimize atmospheric (e.g., aerosols) and radiative geometry effects [38].

The meteorological data were obtained from the Chinese climate academic and science dataset (http://data.cma.cn/, accessed on 2 January 2021), including the month-averaged temperature and month-accumulated precipitation in a period of 1982–2013 from 52 meteorological stations in the Loess Plateau. We further mapped them by inverse distance weight with the boundary and spatial resolution of NDVI imagery.

Vegetation classification maps, from 2001 to 2012, were obtained from MODIS land cover product (MCD12C1) with a spatial resolution of 0.05 degree (5.6 km). We further chose the unchanged vegetation type through Boolean calculation and resampled it with the spatial resolution and boundary of NDVI (Figure 2). Finally, we choose three types of vegetation, Mixed Forest (466 pixels), Grassland (4594 pixels), and Barren or Sparse Vegetation (93 pixels). It is mainly Mixed Forest in the southeast, Barren or Sparse Vegetation in the northwest, and the others are Grassland.

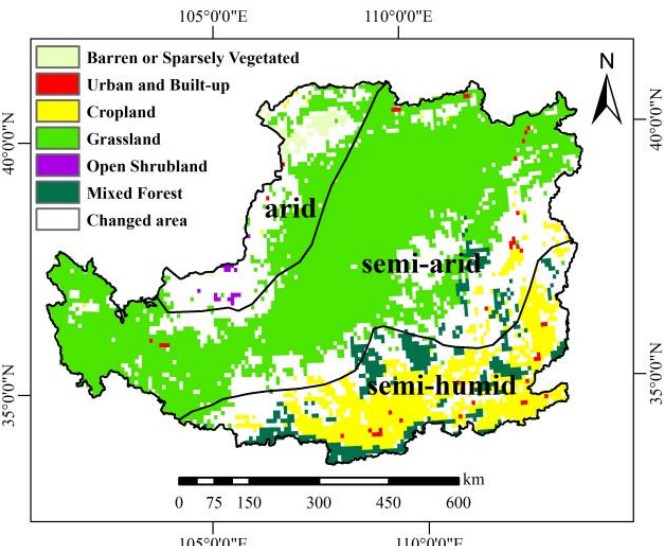

**Figure 2.** Unchanged vegetation map of MCD12C1 product from 2001 to 2012 in the Loess Plateau.

## 3. Proposed Method

To properly understand the proposed method, it is necessary to also understand some of the most known methods in the literature; these methods include the No-lag method, the Lag method, and the Lag-accumulation method. Once these methods have been explained, we will propose a new method, named the weighted time-lag method. Then, different linear regression strategies are adopted to evaluate the superiority of the proposed method.

### 3.1. Previous Lag Methods

There are two knowledge bases of the method to be proposed:

(1) The maximum length of periods in which vegetation respond to climate

Considering the scale dependency [24] of time lag, to provide an example, we compare the differences of methods in detecting the month lag of vegetation response to precipitation and temperature. This monthly lag is detected within three months [25,30]; therefore, we use it as an assumption that the furthest time lag of climate is the third-to-last month, which still impacts vegetation in the current month.

(2) Different periods contribute unevenly to vegetation

Figure 3, as an example, demonstrates the consecutive variation of impact intensity of climate on vegetation [34], where the *x*-axis refers to the time (month), and T is the current month; the *y*-axis refers to impact weights of climate. Curves with different colors refer to the impact variation of the climate in a particular month, where the *y*-value at the intersection (triangles) of the curve and the dashed line of the different months are the impact weights of this climate factor on vegetation. The blue curve, for example, is the variation of the impact weights of a climate factor in (T − 2) month, and it is more influential in (T − 2) and (T − 1) and weaker in (T) and (T + 1).

Vegetation in the current month (T), for example, is impacted by climate factors in months (T − 3), (T − 2), (T − 1), and T (Figure 3), where the *y*-value at the intersection (triangles) of these curves in month T refers to the weights of climate in each month.

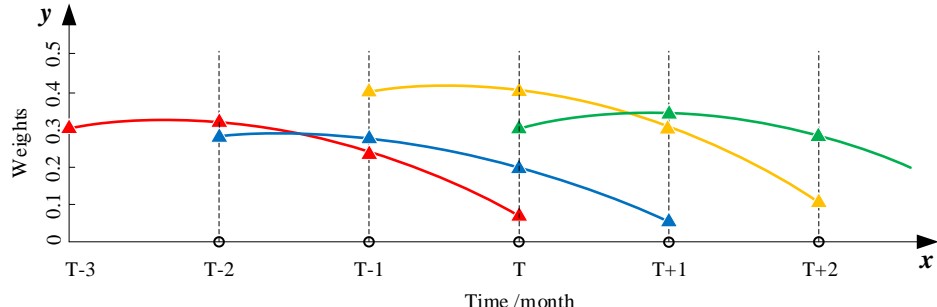

**Figure 3.** Consecutive impacts of climate factors on vegetation in different months. Note: The curves of red, blue, yellow, and green represent the influence of climate factors from (T − 3), (T − 2), (T − 1), and T months to vegetation growth in the following months, respectively. The triangle represents the intersection of the curve and the dashed line. The *y*-value at the intersection represents the influence weights of the climate factors on vegetation in the corresponding month.

We reviewed previous lag methods and categorized different methods based on their characteristics in the following ways.

(1) **No lag method.** Since this method does not consider the time lag, it only uses the climate in month T to detect the vegetation response to climate. From the weight perspective, the weights for the climate in different months, (T − 3), (T − 2), (T − 1), and T, would be 0, 0, 0, and 1, respectively.

(2) **Lag method.** This method assumes the vegetation in T month is only impacted by the climate in the month, which has the maximum of the square of correlation coefficient ($R^2$) in linear regression through input climate in months (T − 3), (T − 2), (T − 1), and T, respectively. In other words, this method assigns the weights to the climate in these months as follows: 1, 0, 0, 0; 0, 1, 0, 0; 0, 0, 1, 0; and 0, 0, 0, 1. Then the weighted mean values of climate are calculated and input into linear regression. Obviously, the lag method is an extension of the no lag method in considering the time lag if there is one.

(3) **Lag-accumulation method**. This method further extends the lag method in considering that the average value of consecutive months may achieve the highest $R^2$ in the regression if any of a particular month cannot. In other words, this method not only considers the impact of time lag in a specific month but also assumes that it could come from consecutive months. For weight, the schemes formulated with this method are shown in Table 1. However, it fails to consider the different impacts of climate in each month (Figure 3) in time series, since the arithmetic means or the sum value of climate factors in the consecutive periods are used as the independent variable in the regression. In other words, their impacts are given equal weight.

**Table 1.** Weight combinations of the lag-accumulation method in different months.

| Schemes | Number of Consecutive Months | Weights | | | |
|---|---|---|---|---|---|
| | | Month (T − 3) | Month (T − 2) | Month (T − 1) | Month T |
| 1 | | 1.0 | 0 | 0 | 0 |
| 2 | | 0 | 1.0 | 0 | 0 |
| 3 | 1 | 0 | 0 | 1.0 | 0 |
| 4 | | 0 | 0 | 0 | 1.0 |
| 5 | | 0 | 0 | 0.5 | 0.5 |
| 6 | 2 | 0 | 0.5 | 0.5 | 0 |
| 7 | | 0.5 | 0.5 | 0 | 0 |
| 8 | | 0 | 0.33 | 0.33 | 0.33 |
| 9 | 3 | 0.33 | 0.33 | 0.33 | 0 |
| 10 | 4 | 0.25 | 0.25 | 0.25 | 0.25 |

### 3.2. Proposed Weighted Time-Lag Method

Considering that the effects of climate factors (e.g., temperature, precipitation) on vegetation are continuity, diversity, and accumulation, a weighted time-lag method is proposed to obtain the weights of the effects of climate factors in different periods on subsequent vegetation growth. We chose different weighting schemes based on the proposed method and constructed a certain number of candidate climate factor series. Then, we calculated the $R^2$ of the candidate climate factor series and NDVI series and selected the highest one as the final $R^2$ of the pixel, and the corresponding weighting scheme of the climate factor series is the final scheme. The process is shown as follows: (1) Assign $P_{T-3}$, $P_{T-2}$, $P_{T-1}$, and $P_T$ as the weights of a climate factor in $(T-3)$, $(T-2)$, $(T-1)$, and T months, where the threshold of them are in [0,1] and $P_{T-3} + P_{T-2} + P_{T-1} + P_T = 1$. (2) Set their sampling interval as SI (SI $\leq$ 1.0). Therefore, the number of probable weight combinations is $M = C_{3+1/SI}^3$; moreover, we obtained a set of corresponding weighted mean values of climate with the number of $M$. (3) Input those weighted mean values in the linear regression with NDVI. (4) The optimal weight combination is achieved by meeting a criterion (maximum $R^2$, for example). Therefore, the detected weights are the corresponding $P_{T-3}$, $P_{T-2}$, $P_{T-1}$, and $P_T$.

The value of SI is set to 0.1 with $M = C_{3+1/0.1}^3 = 286$ probable weight combinations (Table 2). In the following parts, for convenience, we will use methods 1–4 to refer to the no lag, lag, lag-accumulation, and the proposed weighted time-lag methods, respectively.

**Table 2.** The weight combinations of the weighted time-lag method (partial).

| Schemes | Weights | | | | Schemes | Weights | | | | Schemes | Weights | | | |
|---|---|---|---|---|---|---|---|---|---|---|---|---|---|---|
| | T − 3 | T − 2 | T − 1 | T | | T − 3 | T − 2 | T − 1 | T | | T − 3 | T − 2 | T − 1 | T |
| 1 | 0 | 0 | 0 | 1 | 151 | 0.2 | 0.3 | 0.5 | 0 | 277 | 0.8 | 0 | 0 | 0.2 |
| 2 | 0 | 0 | 0.1 | 0.9 | 152 | 0.2 | 0.4 | 0 | 0.4 | 278 | 0.8 | 0 | 0.1 | 0.1 |
| 3 | 0 | 0 | 0.2 | 0.8 | 153 | 0.2 | 0.4 | 0.1 | 0.3 | 279 | 0.8 | 0 | 0.2 | 0 |
| 4 | 0 | 0 | 0.3 | 0.7 | 154 | 0.2 | 0.4 | 0.2 | 0.2 | 280 | 0.8 | 0.1 | 0 | 0.1 |
| 5 | 0 | 0 | 0.4 | 0.6 | 155 | 0.2 | 0.4 | 0.3 | 0.1 | 281 | 0.8 | 0.1 | 0.1 | 0 |
| 6 | 0 | 0 | 0.5 | 0.5 | 156 | 0.2 | 0.4 | 0.4 | 0 | 282 | 0.8 | 0.2 | 0 | 0 |
| 7 | 0 | 0 | 0.6 | 0.4 | 157 | 0.2 | 0.5 | 0 | 0.3 | 283 | 0.9 | 0 | 0 | 0.1 |
| 8 | 0 | 0 | 0.7 | 0.3 | 158 | 0.2 | 0.5 | 0.1 | 0.2 | 284 | 0.9 | 0 | 0.1 | 0 |
| 9 | 0 | 0 | 0.8 | 0.2 | 159 | 0.2 | 0.5 | 0.2 | 0.1 | 285 | 0.9 | 0.1 | 0 | 0 |
| ... | ... | ... | ... | ... | ... | ... | ... | ... | ... | 286 | 1 | 0 | 0 | 0 |

### 3.3. Regression Strategy

We adopt linear regressions to analyze the time-lag patterns of climate factors impacting vegetation, respectively. Considering the different responses of vegetation communities, or even of the same vegetation in a different area, we studied on a pixel scale to reveal the response pattern of vegetation to climate. Furthermore, descriptive statistics were implemented to clarify the different performances of the four methods in linear regression.

We adopted Equation (1) to reveal the different performance of the methods in linear regression and the time lag response of vegetation to precipitation and temperature:

$$\begin{cases} NDVI_{i,j}(k) = a_{i,j}^m(k) \cdot PRE_{i,j}^m(k) + b_{i,j}^m(k) \\ NDVI_{i,j}(k) = c_{i,j}^m(k) \cdot TEM_{i,j}^m(k) + d_{i,j}^m(k) \end{cases} \tag{1}$$

where $NDVI_{i,j}$ refers to the $NDVI$ time series in the growing season (April to October) from 1982–2013 of the $(i, j)$ pixel; $a_{i,j}$ and $b_{i,j}$ refer to the slope and intercept in the NDVI linear regression with precipitation of the $(i, j)$ pixel; $c_{i,j}$ and $d_{i,j}$ denote those with the temperature of the $(i, j)$ pixel; $m$ is the method index from methods 1 to 4; $k$ is the index of a

scheme of a corresponding method; the numbers of schemes of the methods ($m = 1$ to 4) are 1, 4, 10 and 286. For example, when $m = 3$, $k = 1, 2, \ldots, 10$.

To explore the response mechanism of NDVI to climate factors in different growing months (April–October) using method 4, Equation (2) was used for the regression of NDVI with temperature or precipitation in different months.

$$\begin{cases} NDVI_{i,j}^n(k) = a_{i,j}^n(k) \cdot PRE_{i,j}^n(k) + b_{i,j}^n(k) \\ NDVI_{i,j}^n(k) = c_{i,j}^n(k) \cdot TEM_{i,j}^n(k) + d_{i,j}^n(k) \end{cases} \tag{2}$$

where $NDVI_{i,j}^n$ refers to the NDVI time series in the $n$th month from 1982–2013 of the $(i, j)$ pixel; the value of $n$ ranges from 1 to 7, which corresponds to April to October; the meaning of other symbols is similar to that of equation 1.

## 4. Results

Our study uses the long-term series of NDVI and climate factors in linear regression during 1982–2013 on the Loess Plateau. The superiority of the proposed method is proved by comparing the previous methods from the spatial distribution and statistical results of time lag. Based on this, the new method is used to further explore the time lag effect between climate factors and vegetation in different months.

### 4.1. Comparison of the Different Lag Method in Linear Regression

To evaluate the effects of precipitation and temperature on vegetation, we used linear regression for each climate factor of four methods. Figures 4 and 5 are the correlation coefficients ($r$) of NDVI-precipitation and NDVI-temperature. The correlation between NDVI and climate factors is mainly positive, while there are some differences in the southern area in the Loess Plateau, which is mainly dominated by Cropland, which is determined by planting cycles and irrigation. Generally, climate factors and NDVI are less related in method 1, which does not consider the time lag, where those of precipitation and temperature are 0.51 (the average of the absolute correlation coefficients ($|r|$) of all pixels, the same below) and 0.67, respectively. The climate factors formulated by method 2, which consider the time lag to be a certain month, exhibit a higher $|r|$ value with NDVI, 0.53, and 0.75, respectively. Method 3, which extends method 2 under considering consecutive months, enhances the $|r|$ values to 0.61 and 0.78. Method 4 shows the best performance with the highest $|r|$ value among the four methods, where the $|r|$ values are 0.62 and 0.79. The results indicate that our method outperforms other methods, which show the highest correlation between climate factors and vegetation.

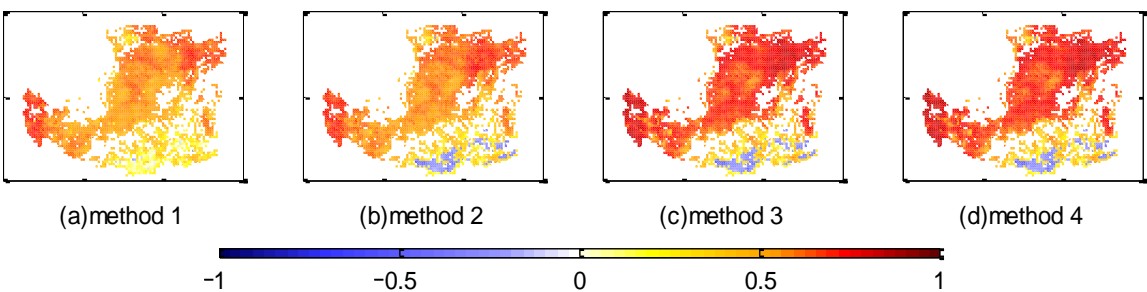

**Figure 4.** The correlation coefficient between normalized difference vegetation index (NDVI) and precipitation was formulated by different time lag methods.

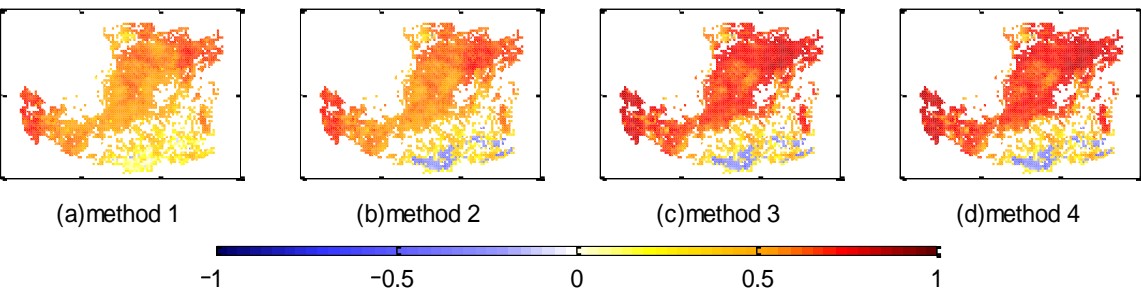

(a)method 1     (b)method 2     (c)method 3     (d)method 4

−1     −0.5     0     0.5     1

**Figure 5.** The correlation coefficient between NDVI and temperature was formulated by different time lag methods.

The time-lag patterns detected by methods 2 to 4 are shown in Figures 6–8. Considering that method 2 only considers the effect of time lag, it can only reveal the time lag of the impact of precipitation and temperature on vegetation. The results are shown in Figure 6a,b, indicating that the time lag of the impact of precipitation and temperature are 0 or 1 month in arid and semi-arid areas, and 3 months in semi-humid areas. In addition to considering time lag, method 3 also considers the accumulation of precipitation and temperature effects on vegetation. Figure 7a,c show the time lag of precipitation and temperature effects, respectively, and Figure 7b,d show the accumulation of those effects, respectively. It can be seen that the time lag of precipitation and temperature are mainly 0 months, while the accumulation is mainly 1 or 2 months. In other words, method 3 reveals that the impact of precipitation and temperature mainly comes from the last 1 or 2 months. In contrast with methods 2 and 3, the proposed method (method 4) can obtain the impact weights of climate factors in the previous three months and the current month on the vegetation of the current month. Method 4 (Figure 8) provides the best weights per pixel for climate factors in different months by the highest $R^2$, which cannot be detected by the other two methods. Figure 8a–d represent the impact weight combinations of $(T − 3)$, $(T − 2)$, $(T − 1)$, and T months' precipitation on T month's NDVI, respectively, and Figure 8e–h represent the weight combinations of temperature, respectively. The weight combinations of the two climate factors are different, but the higher weights of both mainly concentrate in the last two months, where the average of the weights of precipitation and temperature in T month for all pixels are 0.51 and 0.53, respectively, and the average of the weights in $(T − 1)$ month are 0.36 and 0.42, respectively.

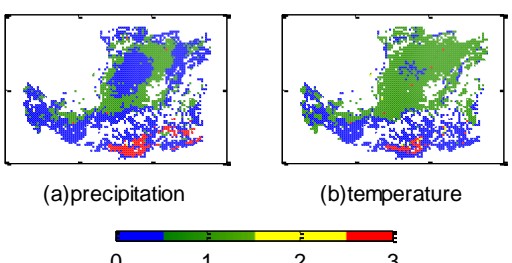

(a)precipitation     (b)temperature

0     1     2     3

**Figure 6.** Time lag of climate impact on vegetation formulated by method 2; 0 represents no time lag, 1 represents one-month lag, 2 represents two-month lag, 3 represents three-month lag.

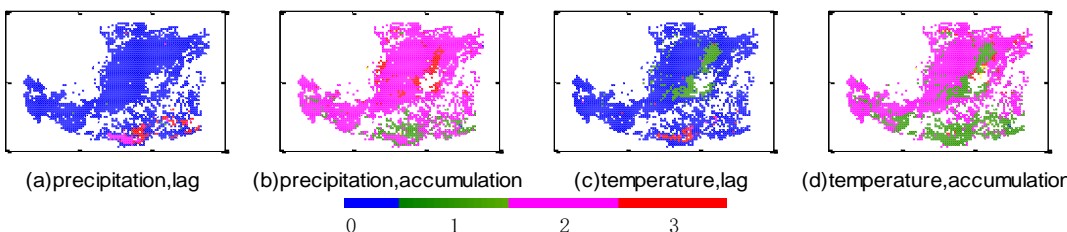

(a)precipitation,lag     (b)precipitation,accumulation     (c)temperature,lag     (d)temperature,accumulation

0     1     2     3

**Figure 7.** Time lag and accumulations of climate impact on vegetation formulated by method 3.

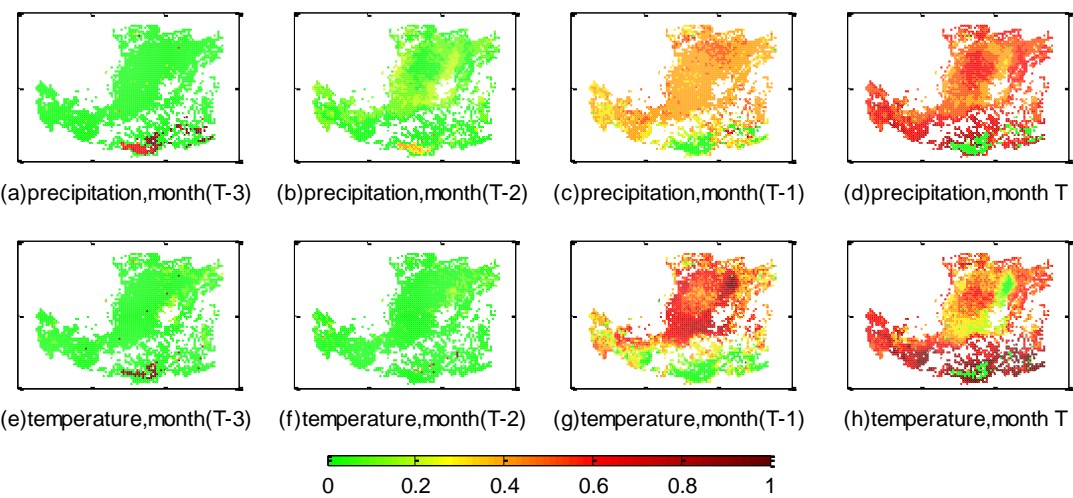

**Figure 8.** Weight combinations of climate impact on vegetation formulated by method 4.

### 4.2. The Statistics of the Results of Linear Regression

Descriptive statistics (Figure 9) are implemented to reveal the time-lag patterns of the four methods in each vegetation community, where the data are the arithmetic mean values of $|r|$ in each method. Three results can be inferred from Figure 9: (1) The $|r|$ values of Grassland with climate factors are the highest in different methods. (2) The $|r|$ values of vegetation with its temperature are higher than those of its precipitation. (3) There is an increase in $|r|$ values from methods 1 to 4.

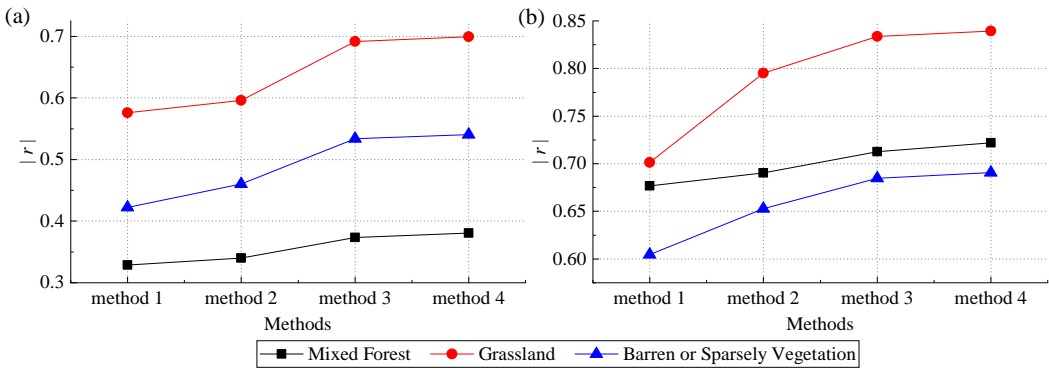

**Figure 9.** The average $|r|$ of NDVI of each vegetation community with precipitation and temperature formulated by each method. (**a**) Precipitation; (**b**) temperature.

The negative *r* suggests in some areas of the south of the study region (Figures 4 and 5), and the corresponding pixels show obvious differences in time lag from those with positive *r* (Figures 6–8). Moreover, the statistic results of method 4, for example, show that the pixels with negative *r* are mainly located in Cropland since the ratios of the number of the pixels with negative *r* in each vegetation (Mixed Forest, Grassland, Cropland, Barren or Sparse Vegetation) to the whole are 7.94% (37 pixels), 0.00% (0 pixels), 28.39% (289 pixels) and 2.15% (2 pixels), respectively, in precipitation; 0.21% (1 pixel), 0.00% (0 pixels), 14.24% (145 pixels), and 0.00% (0 pixels) in temperature.

Figures 10–12 show the average calculated from Figures 6–8. The results are as follows:

(1) Vegetation communities exhibit diverse time lags; however, they generally show the same gradient pattern in lag values detected in methods 2 to 4. This pattern consists of two levels, which are Mixed Forest, Grassland, and Barren or Sparse Vegetation. The corresponding lag values, precipitation in method 2 for example, are 0.13, 0.39, and 0.49 months (Figure 10).

(2) There is an increase in the accuracy of the time-lag responses of vegetation from methods 2 to 4. For example, the time-lag responses of Grassland are around 0.4 months to precipitation and 0.75 months to the temperature detected by method 2 (Figure 10). However, the impacts detected by method 3 come from the last two months for both climate factors (Figure 11). Method 4 (Figure 12) further enhanced the accuracy in showing the impacts of time lag mainly contributed by 52%, 39%, and 8% from the recent three months in precipitation and 48% and 49% from the recent two months in temperature.

(3) The lag values or their contributions to climate factors are similar in each method. The lag impacts of precipitation and temperature to Mixed Forest in method 4 (Figure 12), for example, are 65% and 73%, contributed by the current month and 32% and 26% by last month.

(4) In terms of method 4 (Figure 12), the time lag impacts of precipitation and temperature are mainly contributed by the recent two months, where their contribution proportions are 90% and 97%, but there are some differences in types of vegetation. The contribution proportion from the current month and last month are similar in Grassland and Barren or Sparse Vegetation. However, the impact contribution to Mixed Forest is mainly concentrated in the current month, where it is about 70% for both climate factors, and that of the last month is about 30%.

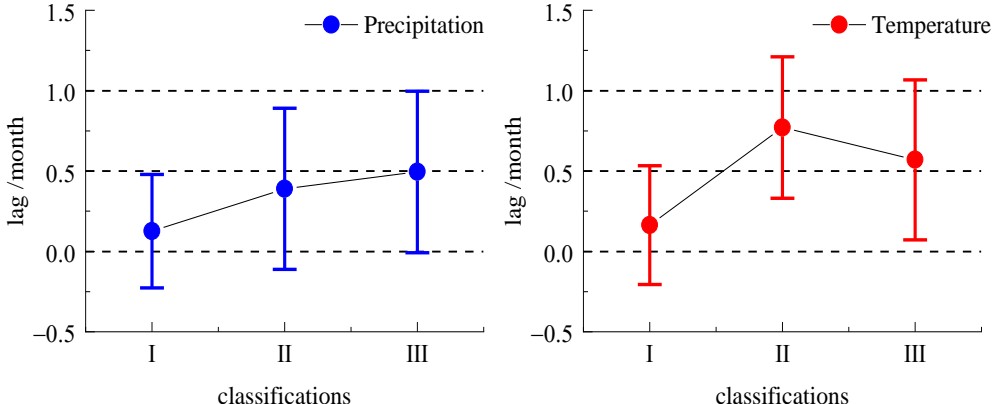

**Figure 10.** Average values (dot) and standard deviations (I-shaped lines) of the lag patterns detected by method 2. I—Mixed Forest; II—Grassland; III—Barren or Sparse Vegetation.

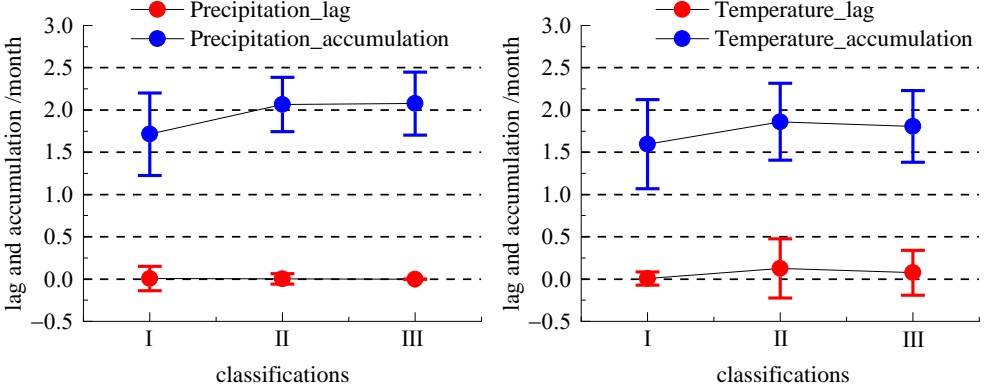

**Figure 11.** Average values (dot) and standard deviations (I-shaped lines) of the lag patterns detected by method 3. I—Mixed Forest; II—Grassland; III—Barren or Sparse Vegetation.

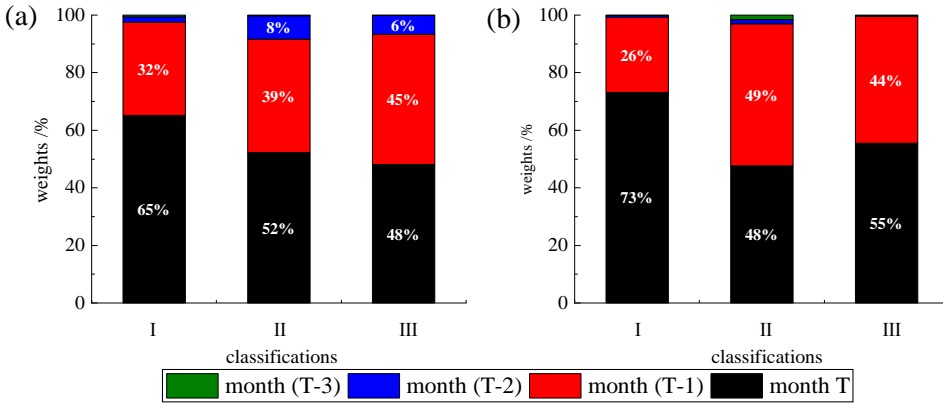

**Figure 12.** Average values of the lag patterns detected by method 4. (**a**) Precipitation; (**b**) temperature; I—Mixed Forest; II—Grassland; III—Barren or Sparse Vegetation.

*4.3. Linear Regression of NDVI and Climate Factors in Different Months Using the Weighted Time-Lag Method*

Method 4 was used to analyze the correlation between NDVI and different climate factors in the monthly time series from 1982 to 2013 in the Loess Plateau. Figures 13 and 14 show the significance level (*p*) of NDVI with precipitation and temperature in different months. The average value of precipitation in the significant areas during the growing season (April to October) was 53.31% (*p* < 0.05) and was greater than temperature, which averaged 39.75% (*p* < 0.05). There were obvious differences in significant areas with the same climate factor and different months. For example, the smallest number of significant pixels was for the impact of precipitation on NDVI in April and May, less than 40% (*p* < 0.05), while the number of significant pixels of impact on NDVI in July to September increased significantly, reaching more than 65% (*p* < 0.05). The number of significant pixels in which temperature had the smallest influence on NDVI in July and August were less than 35% (*p* < 0.05), and the largest in April and May, reaching more than 45% (*p* < 0.05).

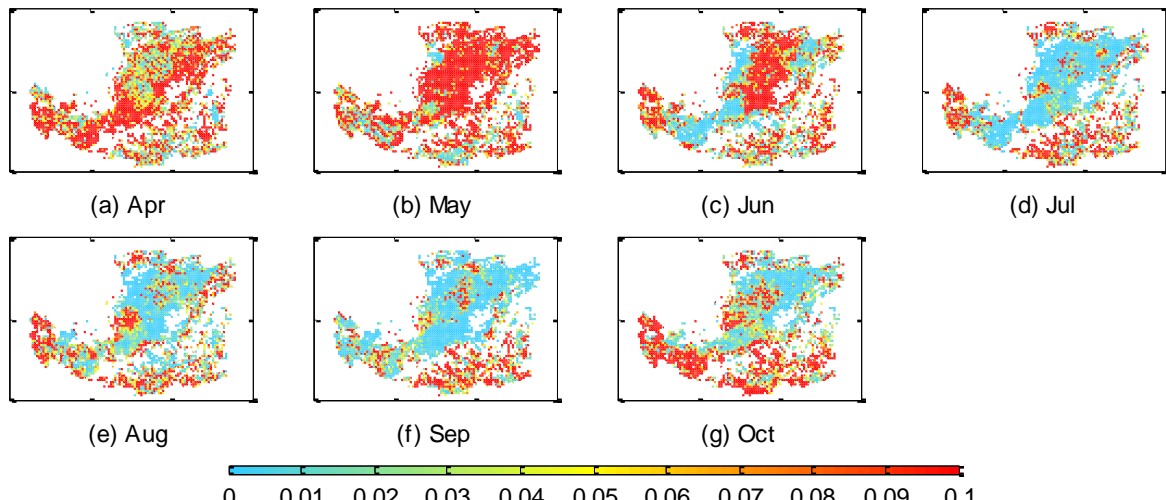

**Figure 13.** The significance level of a linear regression between NDVI and precipitation in different months.

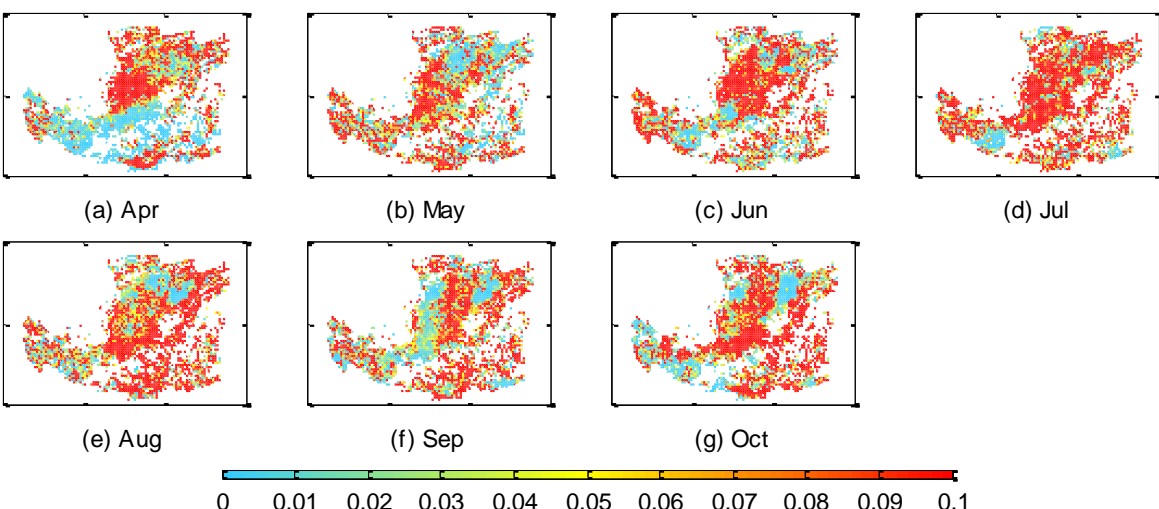

**Figure 14.** The significance level of a linear regression between NDVI and temperature in different months.

Figure 15 shows the proportions of significant pixels between NDVI and climate factors in different months. For different climate factors, positive correlation and negative correlation were different in different months. The influence of precipitation on NDVI in April was mainly negatively correlated, while the influence of NDVI in other months (May to October) was mainly positively correlated. The influence of temperature on NDVI was different from precipitation. Its influence on NDVI in April and May was mainly positively correlated and negatively correlated in other months (June to October).

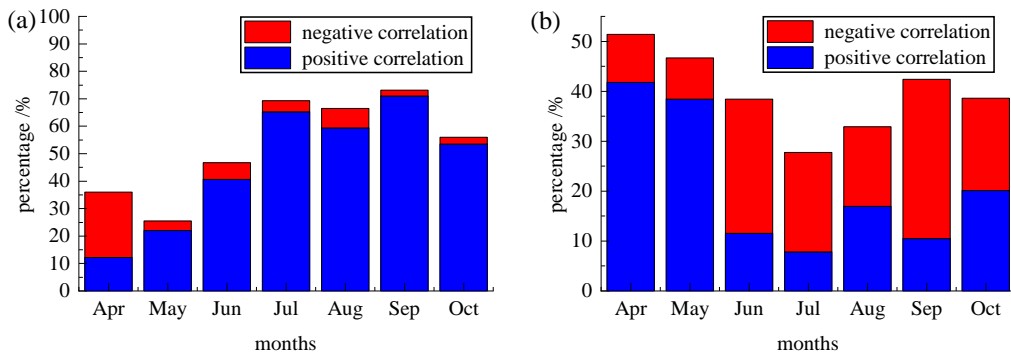

**Figure 15.** Percentage of significant pixels obtained by linear regression of NDVI and precipitation/temperature in different months. (**a**) precipitation; (**b**) temperature.

Different vegetation types have different degrees of response to climate factors. Figure 16 shows that (1) the response characteristics of Mixed Forest to temperature have obvious monthly differences. The number of significant pixels was the largest in April, with a proportion of 87% ($r = 0.62$, $p < 0.05$), and all were positively correlated, while in other months, the proportion of significant pixels decreased significantly, especially in June to September, when the number of significant pixels was relatively small and mainly negatively correlated. The response characteristics of Mixed Forest to precipitation are not obvious. Except for the negative correlation in April, the positive and negative correlations of other months have a certain degree. (2) The response of Grassland to precipitation was significantly larger than that of Mixed Forest. The proportion of significant pixels in July to September reached 70%-85%, and it was mainly positively correlated; the corresponding correlation coefficients are 0.53 ($p < 0.05$), 0.50 ($p < 0.05$), and 0.52 ($p < 0.05$), respectively. However, the significant pixels affected by April and May were smaller, and the correlation between NDVI and precipitation in April was mainly negatively correlated. The response characteristics of Grassland to temperature were significantly different from those of pre-

cipitation. The influence of temperature on NDVI in April and May was mainly positively correlated, while in other months (June to October), it was positively correlated in some pixels, and negatively correlated in some pixels (Figure 13). (3) The response of Barren or Sparse Vegetation to precipitation was the most significant in July to September, and it was mainly negatively correlated in April. The response of Barren or Sparse Vegetation to temperature is generally more negatively correlated than positively correlated, and especially in June.

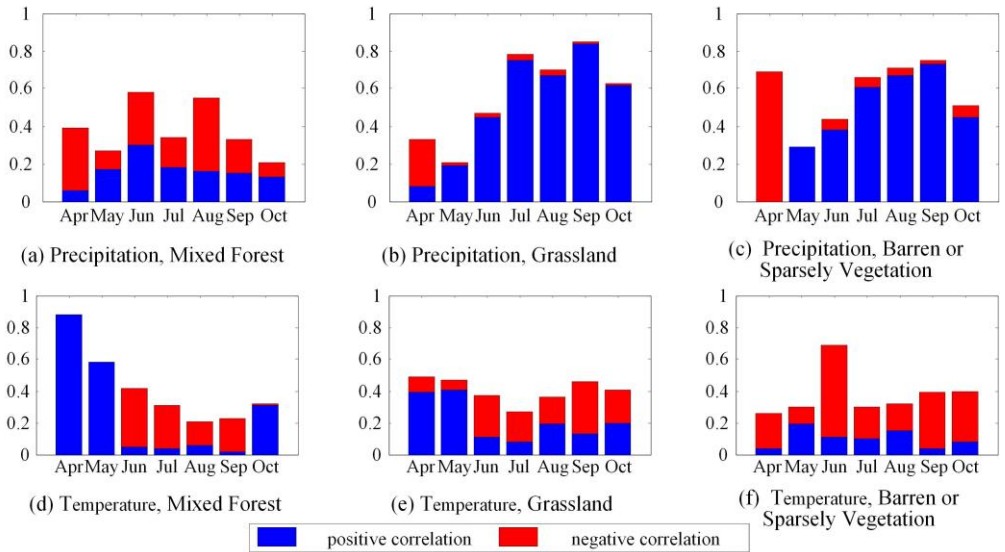

**Figure 16.** The statistical results of the percentage of significant pixels using linear regression of different types of vegetation and precipitation/temperature in different months.

## 5. Discussion

### 5.1. Comparison of Different Time Lag Methods

The result shows that there is a noticeable improvement in fit closeness ($|r|$) in linear regression after considering time lag. This indicates the existence of the impact of time-lag of climate on vegetation, which is also supported by many studies [13,15,30,39–41].

We consider the previous lag methods as two paradigms: lag and lag-accumulation. The lag patterns detected by them differ from each other (Figures 6, 7, 10 and 11). The comparison results show that the $|r|$ achieved by lag-accumulation is higher than that of lag in linear regressions (Figure 9). Moreover, the former one is more accurate in the detected lag patterns; therefore, it is better in revealing the lag mechanism, which is also supported by previous studies [17,18,27,42]. We believe the main reason is that lag accumulation extends the lag in considering the impact may come from consecutive periods, which increases the probability of revealing the mechanism of the time lag.

The new method, the weighted time-lag method, provides the highest $|r|$ in both the regression model among the three time-lag methods (Figures 4, 5 and 9), indicating that it has the highest explanatory of precipitation and temperature on vegetation variation. Moreover, in contrast with the other two, the new method reveals a pattern of the most accurate weight (Figures 8 and 12), indicating that it can detect the impact intensity of climate in each month. Concretely, the lag patterns of precipitation and temperature from months (T − 3) to T are 4.7%, 7.5%, 36.4%, 51.4% and 3.2%, 1.6%, 42.1%, 53.1% (Figure 8). However, these cannot be detected within lag and lag-accumulation methods. This is mainly because lag and lag-accumulation models assume the impact comes from one or more consecutive months, but the new method considers this impact as an accumulation of different weights from previous periods. In other words, we believe the impact of climate on vegetation exhibits accumulation, consecutively and shifting [34,43].

*5.2. Impact of Precipitation and Temperature on Vegetation in Loess Plateau*

The results of simple linear regression show the vegetation dynamics, especially responses to the recent two months, and this response varies in vegetation communities in weight patterns and impact degrees.

The weight patterns of climate in different months shift in the types of vegetation, and the precipitation and temperature align well in the same vegetation (Figures 10–12). Grassland and Mixed Forest, for example, are impacted by temperature in the current month (48% and 73%) and the last month (49% and 26%) detected by method 4. Moreover, the impacts of precipitation in the recent two months are 52% and 39% in Grassland and 65% and 32% in Mixed Forest (Figure 12). Regions with high vegetation coverage have a greater demand for water and heat, so the vegetation in these regions (mainly Mixed Forest) uses the water from precipitation and the heat from higher temperature efficiently within a short span of time, leading to the bigger weight of the influence of temperature and precipitation in the current month [26,30]. This indicates that vegetation communities have differences in response to climate factors; moreover, the precipitation and temperature have a similar pattern in the same vegetation. This is also supported by Wang et al. [17] and Guo et al. [42] in Central Great Plains, USA, and Yalu Tsangpo River Basin, China.

The other finding is that the impact of temperature is higher than that of precipitation in terms of their $|r|$ achieved in simple linear regression with NDVI (Figures 4, 5 and 9). Moreover, the mean values of the $|r|$ are 0.62 and 0.79 for precipitation and temperature (Figures 4 and 5). Xu et al. [44] achieved a result of 0.74 and 0.84 for these two mean values in Inner Mongolia from 1991–2000, which supports this conclusion. For different vegetation, temperature (or precipitation) has different impact weights, where Grassland has the highest $|r|$ as 0.84 (or 0.70), and it is obviously higher than others (Figure 9).

Compared with temperature, the impact of precipitation varies in vegetation communities. Specifically, there is a light impact of precipitation on Mixed Forest ($|r|$ = 0.38), which is obviously lower than that of temperature ($|r|$ = 0.72). This is typically due to the spatial pattern. The Mixed Forest is mainly located in th southeastern Loess Plateau, which is a semi-humid area [36]; therefore, it exhibits lower dependency on precipitation. However, Grassland and Barren or Sparse Vegetation, which are mainly in arid and semi-arid areas, show higher impacts of precipitation ($|r|$ = 0.70, Grassland; $|r|$ = 0.54, Barren or Sparse Vegetation), indicating that vegetation has a strong response to precipitation in arid areas than in humid areas [15,45,46].

*5.3. Lag Effects of Climate Factors on Vegetation on the Loess Plateau*

The influence of climate factors on NDVI has obvious differences in different months of the growing season on the Loess Plateau. The response of vegetation to precipitation is smaller in April and is mainly negatively correlated, while in July to September, the significant area is larger and mainly positively correlated, especially for Grassland; that is, when the temperature is lower in April, precipitation restricts vegetation growth to a certain extent. Considering that the study area is mainly located in arid and semi-arid areas, water resources are limited. As the temperature rises in the middle and late stages of the growing season, precipitation is the main promotion factor of vegetation growth, which is similar to the research results of Mo et al. [47] and Guo et al. [48]. The effect of temperature on vegetation growth is mainly positively correlated in April and May and is negatively correlated from June to September, especially in Mixed Forest. This finding is consistent with the conclusion of Zhou et al. [49]. That is, at the beginning of the growing season, the increase in temperature promotes the growth of forests and other vegetation types, but in the middle of the vegetation growth, the increase in temperature restricts the growth of forests and other vegetation types. Furthermore, since vegetation is highly sensitive to temperature changes at the beginning of the growing season [50,51], taking into account the lag effects, spring warming will reduce frost injury, increase the time for vegetation to turn green, and promote photosynthesis activity [52], thereby promoting vegetation growth. Therefore, temperature is mainly positively correlated with NDVI in April and

May. At the same time, it should be pointed out that the influence of climate change on vegetation growth has complex multi-scale characteristics [24,53], and the experimental verification part of this paper only uses the monthly lag as an example; method 4 has been validated and used, and it needs to be extended to other scales of climate and vegetation growth in time lag analysis.

## 6. Conclusions

In this paper, we summarized the different types of methods to detect the time lag; moreover, we proposed a new weighted method by considering the continuity, shift, and accumulation of the climate impact on vegetation. We further compared our method with others within a case study, where we used them in a study to detect the response of vegetation to the climate in Loess Plateau from 1982–2013. The results address the differences in lag values detected by diverse methods and the advantages of the new method, where the new method can reach the most accurate lag patterns of climate factors.

We further applied the new method to study the response of vegetation in the growing season to the climate in the Loess Plateau. We find that the impact of climate mainly comes from the current and the last months, where the precipitation pattern aligns well that of temperature in the same vegetation, and the weights of precipitation (or temperature) vary in vegetation communities. In addition, the impact of temperature is higher than that of precipitation across the Loess Plateau. The response of vegetation to climate is a complex process, exhibiting spatial heterogeneity; therefore, an adequate model is essential to generate domain knowledge from the study. Our proposed method is just a way to support scholars to understand such a process better. Further research would focus on ecology model improvement and the selection of the parameter estimation approach. Moreover, our method can be still improved, by increasing the amount of the weight combinations increase exponentially with the increase of the independent variables. Therefore, a challenge remains to efficiently find a more accurate weight combination.

**Author Contributions:** Conceptualization, C.L.; methodology, C.L. and Q.S.; software, C.L.; validation, T.C. and A.Z.; investigation, T.C. and Q.S.; resources, Q.S and A.Z.; data curation, C.L. and T.C.; writing—original draft preparation, C.L.; writing—review and editing, T.C. and Q.S.; visualization, A.Z.; project administration, C.L. and Q.S.; funding acquisition, A.Z. All authors have read and agreed to the published version of the manuscript.

**Funding:** This work was partially supported by the National Natural Science Foundation of China (Grant No. 42071246), Natural Science Foundation of Hebei Province (Grant No. E2020402006) and National Natural Science Foundation of China (Grant No. 4207012036).

**Data Availability Statement:** The data that support the findings of this study are available from the corresponding author (C.L.) upon reasonable request.

**Conflicts of Interest:** The authors declare no conflict of interest.

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
