# Peer review of "A Weighted-Time-Lag Method to Detect Lag Vegetation Response to Climate Variation: A Case Study in Loess Plateau, China, 1982–2013"

_remotesensing, doi:10.3390/rs13050923_

Round 1

Reviewer 1 Report

This manuscript has studied the differences between some models representing the process that ecosystems respond to the climate with a time lag. Overall the paper is good, but there are some comments to do.

MAJOR COMMENT: the paper needs to have some descriptions about the reason for the different lag time between different parts of the study area.

Introduction:

  • This part needs to add some newest studies in the field of NDVI variations in relation to climatological factors.
  • Lines 88-97 could be moved to METHODOLOGY or first of DISCUSSION sections.

Materials and Methods

       - Figure 1: Having the elevation of the study area (in shaded) can be informative. Add it, please.

- Lines 114-119 should have a reference.

- Lines 126-127. Add a sentence about the advantages of using maximum-value composite (MVC).

- Lines 134-135. Why you have used the Vegetation classification maps, from MODIS land cover product (MCD12C1) with the spatial resolution 0.05 degree while there is MCD12Q1 images with 500 meters spatial resolutions?

- LINES 136. What are the advantages of Boolean calculation? Should be addressed.

- Lines 137-138. Why you have chosen LC types, which include more than 50 pixels? Write a bit more about why your choice (50 pixels).

Discussion:

Maybe adding some information about the local indicators (elevation, or other local factors) that affect the lag time will be helpful. See Figure 8 and the southern part of the figures. Try to add some reasons about why the southern part is really unique?

Figure 9, 10, 11: the authors should mention some reasons for the differences in lag time between the land cover types. For example, why the croplands has the most lag time than others? Why the forests have the least lag time and so on … .

Author Response

We have completed the modification following your comments. The review reports have been uploaded to the system. Please see the attachment. Thank you for your review.

Reviewer 2 Report

The manuscript presented relevant issues and comparison between the different 
results drawn from different lag models. 

The paper is well structured. The introduction section includes all significant related information, sufficiently highlighting the importance of the topic and the novelty of this work. All parts of the methodology are described in a comprehensive manner. The results are valuable and are properly discussed.

The merit is then unquestionably essential.

Nonetheless, still some minor reviews should be performed, such as:

Line 234: Correct "climte"

Line 318: Correct "paradims"

Author Response

We have completed the modification following your comments. The review reports have been uploaded to the system. Please see the attachment (Response to Reviewer#2). Thank you for your review.

Reviewer 3 Report

This manuscript claims to present a new methodology to detect time-lag on vegetation response to climate variation. To do so a case study area is considered located in Loess Plateau un China and a claimed period of analysis is 1982-2013, though it is not actually clear which year are they really analyzing, but for sure only few months of a single year are studied. The so called proposed method is based on weighted-time-lag. Other methods are also presented in the manuscript (4 in total), but from the names or presentation itself it is not possible to understand if there is any proposed one, though authors finally seem to claim that method 4 is the one proposed by them. There is no expalanation at all of the methodology. State of the art analysis is poor to non existent, and actually references are rather old compared to the current year in which we are (2021 vs 2017). English is also problematic in several points. The results are just a mix of several things that do not lead to any proper conclusion and in fact the discussion/analysis is not coherent because of this. Conclusions cannot be really achieved from the current manuscript. A lot of work is still required from authors before considering this manuscript for publication. Further details are provided in the next.

ABSTRACT

- line 28- Just a curiosity, why only NDVI?. If you are talking about influence of climate variation, what about studying other indices that take into account particular climate variations. For instance about the ENSO phenomenon that does affect the vegetation as well, you could study the ONI index, that allows to measure when the phenomena happens.

- line 29. "precipitation and temperature". In fact, for instance the ENSO phenomena does affect both precipitation and temperature according to the phase in which it is.

- line 35. Can the temperature and precipitation along be considered as climate phenomena in here?. Or what do you mean by it?.

- line 38. In fact bought drought and flooding are also consequences of ENSO phenomena. What sort of phenomena are you exactly analyzing?.

INTRODUCTION

- lines 42-43. it was assumed - it has been assumed

- line 50-63. Please consider the next paper as well: Y. T. Solano-Correa, L. Pencue-Fierro, and A. Figueroa-Casas, “Determining the effects of ENSO phenomena on Andean areas by applying radiometric indices on long time series,” in 2015 8th International Workshop on the Analysis of Multitemporal Remote Sensing Images (Multi-Temp), 2015, pp. 1–4.

- line 86. Please add reference(s).

- line 87. English.

- line 92. contribute instead of contributes.

- lines 106-109. So you do not have actual ground truth information in order to test the method proposed?. Furthermore, are you really proposing a new method or simply comparing several existing methods in literature in order to know which one performs better?.

- General: The introduction lacks from a proper problem statement analysis, as well as from a better state of the art analysis and discussion. It is not really clear from the introduction what is the final goal from authors, neither what are they actually proposing or doing. Please improve and revise for minor English typos/problems.

MATERIALS AND METHODS

- line 112. Of which country?.

- lines 113-114. Cause by some particular phenomena?.

- line 126. Could you please provide the spatial resolution in meters. This is a more standard measurement unit in remote sensing.

- line 127. Atmospheric instead of atmosphere.

- lines 127-128. Can you add some reference or further explain this step?. And does this mean that you have 12 images per year in the whole 1982-2013 period?.

- line 134. What about classification maps for the rest of the years that you are analyzing?.

- lines 134-135. How do you put the MODIS classification maps at the same spatial resolution as your datasets?

- line 136. Boolean calculation: can you provide both a reference and also further details on this method?. What about the accuracy of it?. Is it unsupervised?.

- line 137. Considering your low spatial resolution, together with the small size of crop fields in agricultural areas, is it really possible to analyze the time-lag influence of climate variables on such type of vegetation land cover?.

- Figure 2. Please improve caption and description of the figure on the main text. Otherwise it loses relevance.

- line 153. In the case of cropland, also different types of crops may contribute unevenly. How do you take this facto into consideration?

- line 167. So you are actually not proposing a new method, but comparing existing ones?. From your title, it seems like you are proposing a totally new method.

- line 169. Why do you call this type of method as "original"?. It sounds like this is a proposed new method, or is it?.

- line 188. Please move Figure 3 just after it is mentioned.

- line 191. Similar to the "original", why "new"?

- line 205. This part comes a little bit out of no-where. It would be good if you could dedicate some time at the very beginning of the methodology section in order to provide a brief summary of the whole method. In this way it is easier to then follow up the steps provided in here.

- line 208. "on a pixel scale". How does the low spatial resolution of your data impact on the final result?. There is a problem of mixing spectral information in here.

- General: There is a lack of a clear methodology. Even if the goal is that to compare existing ones, there should be a more detailed explanation on how each of them work. Consider that you are sending this paper to a remote sensing journal, which is quiet general. And not to climate based one, which maybe is more aware of these methodologies. Therefore, the chances of you finding possible readers without a proper background are high.

RESULTS

- line 223. So you are really really working with comparison. Please update title, abstract and introduction in consequence and make it clear from the very beginning.

- lines 224-234. And where is a proper explanation on what exactly are you comparing in here?. You go directly to talk about Std, without having even introduced what the term is and without having even talked about the experiments you are performing, or what data, how, is it used. Please consider re-organizing the whole results section, as well as the methodology one, in order to be both coherent and rational on things that you are presenting. Otherwise, there is no added value on this manuscript.

- line 233. "our method". And what is your method?. There was no presentation of any new method in your methodology section, at least no on a explicit way. Do you or don't you have a real new method in here?. Do you see the lack of proper organization and presentation of your manuscript?.

- line 234. climte -> climate.

- Figures 4 and 5. And how is the correlation related to the proper time-lag detection?. Where is the time variable considered in here?. Is correlation to temperature higher than that to precipitation?. I think this is the only thing I can conclude from these two figures. You need to better reformulate your results so far.

- line 242. "appears"?

- line 248. Shouldn't there be at least some coherence along the 4 methods?. How can you claim that method 4 is correct and not the others?.

- Figure 6. Are the colors suppose to represent number of months?. If so, please specify it on the caption.

- Figures 6 to 8. How can one, as a reader, relate all the results presented in here?. They are all totally different from amount and quality. So how can one be sure that what you are presenting is correct/coherent or not?. Is there any possibility to compare to any other existing method that generates some similar products?. Or better to say, can method 4 generate any of the same products as the other methods, thus allowing for a real one to one comparison?.

- line 259. is -> are.

- lines 268-270. Why do you come back to some qualitative results in here, when it is clear from Figure 9, that there are no negative values at all?. This is totally confusing.

- lines 270-277. This whole part is coming out of no-where. Why to talk about these results in here and not in the previous sub-section?. And what is the actual point of this analysis and further sub-division of croplands?.

- lines 285-292. What do you exactly mean by all this analysis?. There is not even coherence in the information shown in Figures 10-12, and they are suppose to be showing the same sort of information.

- line 294. You talk in here about Mixed Forest, whereas on Figure 12 you talk only about Forest as a general. Is there any difference in here?.

- lines 297-303. How exactly do you analyze the information provided in Figure 12?. And what is the point of this analysis?. Aren't you suppose to be comparing accuracy of different methods?. Where is such comparison, and how do you guarantee an even comparison?

- Figure 10. Please add the whole name to variables as Precipitation and Temperature. Unless you pre-define PRE and TEM as their corresponding on the main text and figure caption.

- Figure 12. What is this figure actually showing. It contains information about weights, classification, classes and then you talk on caption about median values.

DISCUSSION

- line 314. "appears"?

- lines 318-325. And how is this supported by your previous results?. I do not see it clearly.

- line 326. "in both". But you are only referring to one thing in the next. Furthermore, I did not see any proper/common comparison among the 4 methods presented on your manuscript.

- line 335. "the new method claims". I am still not sure if this method (number 4) is a proposed one by authors or simply an extra one in literature, but for sure there is no claim as per what is exactly expected from it. So I would not put such words in here.

- line 338. "Climate" what exact climate change phenomena is affecting temperature and precipitation in the analyzed period?. Furthermore, there is not even a single analysis over the 1982-2013 period, not even a mention on different years in here.

- line 344-345. "repose"?

- lines 345-347. There was not even a real comparison among the 4 methods. So what problem would be there for real?.

- line 349. Once again, what do you mean by "appear" in here?. Did you mean "show"?.

- line 350. "responses"?

- lines 371 and 372. What are the units of the precipitation impacts on parenthesis?, and how did you obtain them?.

- line 372. Which are the values for temperature?. You have not dicussed this before.

CONCLUSIONS

- line 383. What exactly is your new method?. It is not actually clear at all from your manuscript so far. There is a whole confusion in here. Furthermore, what does your method do, and how is it better than state of the art methods?. There is no real comparison on the whole manuscript that can really tell the reader that there is a method performing better than others. There is no validation data or grounf truth at all.

- line 386. Other than mentioning on the data description the period to be studied, there is no other mention to the time rather than just 3 months before or after a given time. There are not even maps supporting such analysis.

- line 388. How do you support this claim?.

- lines 389-390. What identified areas?. I did not see any of this.

- lines 399-401. I would be glad if at least your so called "proposed method" would be explained in a proper way. Then of course, a proper experimental and results section would be more than welcome, together with a fair and proper validation part.

REFERENCES

- References need to be totally revised in order to consider recent research, the earliest paper that have cited is only from 2017, and it is just one. It is clear that your literature review is poor and this is maybe reflected on the quality of your current manuscript. Please update and revise again for more recent literature. I do not thing there has not been any research in this same direction over the last 4 years.

Author Response

We have completed the modification following your comments. The review reports have been uploaded to the system. Please see the attachment (Response to Reviewer#3). Thank you for your review.

Reviewer 4 Report

Summary: This paper proposes a weighted lag approach to explain current month NDVI using prior months (0 to -3) precipitation and temperature. The analysis is of high interest to me, I frequently find it necessary to make decisions on how to include climate (lagged, accumulated, current month) in vegetation analyses and the literature is still evolving on this topic. The analysis was at times difficult to follow and made some assumptions that seemed potentially suspect, but in general I think it is a worthwhile manuscript.

General: Grammatical mistakes throughout, please have the paper reviewed by a native English speaker.

Major Comments

Line 148-152 – This first assumption doesn’t seem very valid to me. In the introduction the authors points at Bunting et al. [20] where woodlands showed a lagged response 6-12 months out and grasslands showed a lagged response of 3-6 months, yet the analysis only considers months 0 to -3. The premise of weighted lagged responses seems very intuitive to me but minimally tested in the analysis. For example, in past efforts I’ve seen vegetation responding to both near recent climate (past month or 2) as well as longer term (how much precipitation occurred last year). Why consider such a short amount of time for this analysis?

Line 154-161 – I found Figure 3 super confusing, but wouldn’t we also potentially expect that the relative lag influences will depend on the month of observation? So the beginning of the growing season might show a different weighted lag relative to the end of the growing season? For example in one of my own projects we found that the vegetation was responding most strongly to that years spring precipitation, regardless of what month the vegetation was looked at. This analysis could be easily added by looking at how the weighted lag potentially changes between beginning of growing season (April-May), peak growing season (June-August) and end of growing season (Sept-Oct).

Comment - (4) the new weighted method – this approach of testing all 286 probable weights seems quite inefficient…why not just let a statistical model such as a linear regression or something similar that would produce coefficients for each climate lagged month, figure out what the weights should be? Also, this approach would become exponentially more intensive if others chose to include more lag months (e.g., T-5, T-6)

Comment – The analysis relies primarily on correlation coefficients but these correlations still have assumptions of independence. Given that it is unlikely that a pixels July NDVI is independent from a pixels June NDVI, how are the authors compensating either in the sampling or the statistics to meet these independence assumptions?

Minor Comments

Line 54 – Consider changing to, “different vegetation shows different lag response times to global temperature.”

Line 60 – change to “grassland appears to have a shorter response…”

Line 66 – change to “summarize” from “summary”

Line 74-75 – Looks like this sentence was supposed to be deleted prior to submission?

Line 91-97 - Revise grammar in the research questions, there are multiple errors here. Also, consider re-wording as “how much would they impact” and “how much would they contribute” is quite vague as currently worded. Perhaps, (1) What is the relative contribution of previous months climate to explain vegetation in a current month? (2) Will a weighted approach to characterize lagged climate outperform an arithmetic means or summed lag approach?

Section 2.1 – what is the growing season? and does it vary across the study area?

Line 138 – capitalize “grassland” to be consistent with other cover types.

Figure 2 – change scale to km

Figure 3 – Expand the figure caption to better explain the figure. Right now the explanation including the legend defining the triangles are in the text, not the figure or caption. Also I am having a hard time understanding this figure. The triangles represent the intercepts of what? And why are the yellow and green triangle lines dependent on future conditions?

Table 1 – It looks like this might be showing the schemes tested for the original method (scheme 1), the lag method (schemes 2-4) and the lag-accumulation method(schemes 5-10)? If this is the case change the reference to the table and add a columns that indicate what method the weights are showing.

Section 2.3.2 – So a predicted NDVI was calculated for each index-scheme within each method and the predicted NDVI were then correlated with the actual NDVI values (April-October, 1982-2013) and the correlation values compared to evaluate performance? Clarifying this section would be helpful, it is pretty confusing as currently written and is a key part of the Methods section.

Figures 4 and 5 – besides method 1, the other approaches are testing multiple weights for the method, so was this the averaged or best performing one shown? Please clarify in the text.  And was “best” if that’s what it is, defined as across the entire study area? or, as suggested by Figure 12, perhaps by cover class?

Figure 8 – ok so I’m guessing that this is showing the “best” weights of those tested? Is it the best overall or the best per pixel? Please clarify in the text.

Comment - The performance is pretty poor for the ag areas in the south portion of the study area, this is of course because the timing of the green-up is determined by planting cycles and irrigation, not just climate, would it make sense to exclude cropland from consideration? I was also curious how valid it was to include sparse-non-vegetated…the NDVI must be pretty low in these areas given the assigned land cover…focusing on grassland and forest could allow for an expansion of the analysis as suggested in the above Major comments section.

Author Response

We have completed the modification following your comments. The review reports have been uploaded to the system. Please see the attachment (Response to Reviewer#4). Thank you for your review.

Round 2

Reviewer 1 Report

Greetings, the authors have addressed all of my comments properly and I think the manuscript can be accepted in the current form.

Author Response

感谢您对我们工作的认可。我们非常感谢您在手稿上所做的工作。

Reviewer 3 Report

Authors have replied to most of my comments and implemented only some of the things suggested. On the other hand, most of the times authors have said that they would update the manuscript as suggested, but instead they have not done so. My main concern continues to be the fact that looking only at the manuscript it is not clear what the proposed approach is. In fact, authors had repetedly said that Method 4 in manuscript is their proposed approach, but they have not even updated the document as to express this. And it must be done. A better organization of the whole document is still missing to guarantee coherence. References have been improved but state of the art analysis on the introduction still needs to be updated. Some further comments are below.

- line 115-116. Regarding the spatial resolution, even though it migh be used as well in degrees, it is mostly used in meters. Please add the meters spatial resolution as well, you can keep both if you wish, but this is a rather simple requirement that I do not think authors can deny.

- line 134. Contrary to your answer in my comments, you have not added any short and general description of the methodology in here. You went directly to present some methods without any partiular explanation. This is one of the reasons why it is rather difficult (some times impossible) to follow your exact ideas and to understand what are you really proposing.

- line 191.If, as you said on your answers to my comments, your proposed method is this one. I strongly suggest that you move it out of this list and put it in a separate section where you properly explain all the details of it and name the section with a keyword as "proposed". Otherwise, it is impossible to get that this is your proposed method by simply reading "new".

- line 228. Similar to the methods case, you need a short introduction in here in order to better explain how the results are presented. Even though you have repetly replied on my comment that you have changed the methods and results configuration in order to properly reflect what you are proposing, plus a logical flow, there is no such a change in the updated manuscript.

- Fig. 6, 7 and 8. You replied to me in the comments, but did not add any further explanation on the manuscript that could allow for a better understanding.

Author Response

We have completed the modification following your comments. The review reports have been uploaded to the system. Please see the attachment (Response to Reviewer#3 (Round 2)). We are very grateful for your work on our manuscript.

Reviewer 4 Report

Comment – I like the addition of the monthly analysis.

Figure 1 caption – changing from “included” to “including”

Line 323-325 – This sentence does not make sense as currently written, please revise.

Line 325 – change to “Figures 13 and 14 show the significance level (P)…”

Line 327 – change to “The average value of precipitation in the significant areas during the growing season (April to October) was 53.31% (P<0.05) and was greater than temperature, which averaged 39.75% (P<0.05).”

Author Response

We have completed the modification following your comments. The review reports have been uploaded to the system. Please see the attachment (Response to Reviewer#4 (Round 2)). We are very grateful for your work on our manuscript.

Round 3

Reviewer 3 Report

Thank you for the update of the manuscript, please take into account this final comments:

- Divide section 2 into two sections, one being the "Study are and Datasets", this will be your new section 2, and the new section 3 being "Proposed method". Otherwise things are mixed up. Specially since you have named sub-section 2.3 as "Methods" again.

- With the new suggested division, your previous methods will be part of sub-section 3.1, and the proposed one will be on 3.2. Please rename the sub-section as "Proposed weighted time-lag method".

- When I suggested to add a small and general introduction at the beginning of the whole methods and results sections, it was not about what you have understood. This is more like a short summary of what readers will find next. For example for the method: "In order to properly understand the proposed approach, it is neccesary to also understand some of the most known ones in literature, those are: No lag method, Lag method and Lag-accumulation method. Once those methods have been explained, we will introduce our proposed approach which is based on .... In order to evaluate the reliability of the proposed approach a regression strategy is then used that... ". Same applies for results, where you explain in a general way the way in which results and experiments are carried out and presented. You cannot just go directly to the results, without explaining the way you are presenting them. In particular, in your case it is rather difficult to follow the results, since you have not present an "experiments" section or anything like that. Please add such introductory paragraphs at the beginning of your section 3 (methods) and section 4 (results). Take into account I am using the section numbers updated as suggested for more organization.

Author Response

We have completed the modification following your comments. The review reports have been uploaded to the system. Please see the attachment (Response to Reviewer#3 (Round 3)). We are very grateful for your work on our manuscript.
